# Impaired recruitment of dopamine neurons during working memory in mice with striatal D2 receptor overexpression

Sevil Duvarci[1], Eleanor H. Simpson [2,3], Gaby Schneider[4], Eric R. Kandel[3,5,6,7,8], Jochen Roeper[1] & Torfi Sigurdsson [1]

The dopamine (DA) system plays a major role in cognitive functions through its interactions with several brain regions including the prefrontal cortex (PFC). Conversely, disturbances in the DA system contribute to cognitive deficits in psychiatric diseases, yet exactly how they do so remains poorly understood. Here we show, using mice with disease-relevant alterations in DA signaling (D2R-OE mice), that deficits in working memory (WM) are associated with impairments in the WM-dependent firing patterns of DA neurons in the ventral tegmental area (VTA). The WM-dependent phase-locking of DA neurons to 4 Hz VTA-PFC oscillations is absent in D2R-OE mice and VTA-PFC synchrony deficits scale with their WM impairments. We also find reduced 4 Hz synchrony between VTA DA neurons and selective impairments in their representation of WM demand. These results identify how altered DA neuron activity —at the level of long-range network activity and task-related firing patterns—may underlie cognitive impairments.

[1] Institute of Neurophysiology, Neuroscience Center, Goethe University, Frankfurt, Germany. [2] New York State Psychiatric Institute, New York, NY, USA. [3] Department of Psychiatry, Columbia University, New York, NY, USA. [4] Institute of Mathematics, Goethe University, Frankfurt, Germany. [5] Department of Neuroscience, Columbia University, New York, NY, USA. [6] Kavli Institute for Brain Science, Columbia University, New York, NY, USA. [7] Mortimer B. Zuckerman Mind Brain Behavior Institute, Columbia University, New York, NY, USA. [8] Howard Hughes Medical Institute, New York, NY, USA. These authors contributed equally: Jochen Roeper, Torfi Sigurdsson. Correspondence and requests for materials should be addressed to S.D. (email: duvarci@med.uni-frankfurt.de) or to T.S. (email: sigurdsson@em.uni-frankfurt.de)

The ability of neural circuits to adaptively control behavior is strongly influenced by the neuromodulator dopamine (DA), which is released by neurons residing in midbrain nuclei including the ventral tegmental area (VTA). Much of the research on VTA DA neurons has examined how they detect motivationally salient stimuli such as rewards and support reward-based learning[1–3]. In addition, there is also strong evidence suggesting that DA plays a key role in cognitive processes, most notably working memory (WM)[4,5] although exactly how these processes are supported by the activity of DA neurons is only beginning to be investigated[6,7]. DA neurons can influence cognitive processing by modulating activity within the prefrontal cortex (PFC), which is one of their main projection targets[8] and is critical for many higher-order cognitive functions. Consistent with this possibility, DA receptor signaling influences task-related firing patterns in the PFC[9–11] as well as PFC-dependent cognitive functions such as WM[12,13]. In turn, the PFC sends projections back to the VTA and influences the activity of VTA DA neurons[14,15]. These mutual interactions between the VTA and PFC are also observed in the coordination of neural activity between the two structures[16], which is mediated by 4 Hz oscillations during cognitive tasks in rats[7]. However, the contribution of VTA-PFC synchrony to cognition and behavior, under both normal and pathological conditions, is not fully understood.

In addition to having a role in guiding normal behavior, dysfunction of DA neurons has also been implicated in the cognitive deficits associated with psychiatric illnesses[17]. Altered DA receptor mediated signaling in the PFC is thought to contribute to cognitive impairments, including impairments in working memory, in patients with schizophrenia[18,19]. Furthermore, the detrimental effects of stress on cognitive performance result in part from excess DA release in PFC[20]. These results suggest that disturbances in the electrical activity of DA neurons—including their coordination with activity in other structures such as the PFC—could contribute to cognitive dysfunction. Indeed, several studies suggest that impaired synchrony between the PFC and other brain regions contributes to cognitive deficits in animal models of psychiatric illness[21,22]. However, whether impairments in long-range synchrony between DA neurons and the PFC are also associated with cognitive dysfunction in animal models has not been investigated.

To address these questions, we recorded from pharmacologically identified VTA DA neurons in mice overexpressing the D2-type DA receptor in the striatum (D2R-OE mice)[23] during WM performance. D2R-OE mice model the increase in striatal D2 receptors seen in schizophrenia patients[24] and display some of the cognitive impairments observed in the disease, including deficits in WM[23]. At the neurobiological level, we recently found that these mice display selective alterations in the in vivo firing patterns of DA neurons in the VTA, but not in the substantia nigra, in anesthetized mice[25] and in the DA modulation of synaptic transmission in PFC in vitro[26]. Here we show that the recruitment of 4 Hz synchrony within and between the VTA and PFC during WM is selectively disrupted in D2R-OE mice and that this impairment correlates strongly with their working memory deficits. At the cellular level, this synchrony deficit is selective to DA neurons in the VTA and is also associated with impaired coordination of 4 Hz firing patterns between VTA neurons. Finally, we show that the representation of WM demand, but not reward or choice, by VTA DA and PFC neurons is selectively impaired in D2R-OE mice, suggesting a deficit in a specific neuronal subpopulation. Taken together, these findings are the first to define the signatures of altered DA neuron electrical activity—both in terms of their task-related firing patterns and long-range synchrony—which could underlie cognitive dysfunction.

## Results

**Impaired WM-dependent 4 Hz oscillations in D2R-OE mice.** In order to examine whether alterations in the firing patterns and long-range synchrony of VTA neurons are associated with cognitive deficits, we recorded neural activity simultaneously in the VTA and medial PFC (Supplementary Fig. 1) of D2R-OE and control mice while they performed a delayed non-match to sample spatial WM task in a T-maze (Fig. 1a)[21]. Each trial of the task consisted of two phases. In the "sample phase", mice ran up the center arm of the maze and entered one of the goal arms to collect reward, while the other arm was blocked. After a brief delay, the mice again ran up the center arm, but now both goal arms were open ("choice phase"). To obtain a reward, animals had to enter the goal arm not visited during the sample phase. Consistent with our previous findings[23], D2R-OE mice required more training days to reach criterion performance on the WM task (Fig. 1b, c). This impairment does not reflect a general deficit in appetitive learning or spatial navigation since D2R-OE mice learn a spatial reference task in a T-maze (where one arm of the maze is always rewarded) as well as controls[23]. For subsequent analysis of neural data, we first examined only sessions after animals had reached criterion performance on the WM task in order to ensure that differences in task performance did not influence the results. Indeed, during these sessions, the choice accuracy of D2R-OE and control mice was comparable (Fig. 1d).

In order to examine neural mechanisms underlying the impaired WM performance in D2R-OE mice, we compared neural activity in the sample and the choice phases of the WM task, specifically while animals ran up the center arm of the maze

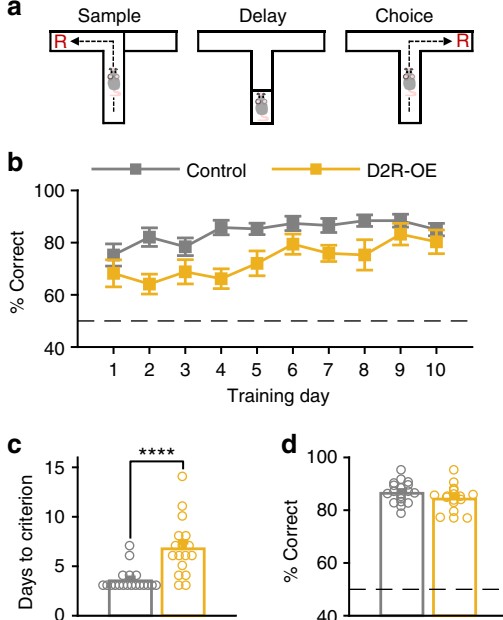

**Fig. 1** D2R-OE mice are impaired on a spatial working memory task. **a** Schematic of working memory task. During the "sample" phase animals were directed to enter one of the goal arms of the maze. After a brief delay period, in the "choice" phase they had to enter the opposite arm to obtain a reward (R). **b** % correct choices during each day of training in D2R-OE and control mice. **c** Days required to reach criterion performance (≥70% correct choices over three consecutive days) in the two groups. D2R-OE mice required more days of training (****$p < .0001$, $n = 19$ control mice and $n = 17$ D2R-OE mice, rank-sum test). **d** % correct choices during testing sessions (after reaching criterion) from which electrophysiological data was analyzed did not differ between genotypes ($p = 0.1364$, rank-sum test). Error bars represent mean ± s.e.m over animals

prior to turning left or right (Fig. 2a). Although the animals' motor behavior was the same in these epochs, WM-guided decision making was only required in the choice phase. We observed that during the choice phase a low-frequency oscillation between 3 and 6 Hz was present in the local field potentials (LFPs) recorded in the VTA of control animals (Fig. 2b, c; hereafter simply referred to as the "4 Hz oscillation"). Accordingly, average LFP power in the 4 Hz range was higher in the choice than in the sample phase in control animals (Fig. 2c, e). This increase in power was not due to differences in running speed, which was similar in the two phases (Supplementary Fig. 2b). In striking contrast to control animals, this choice-phase selective increase in 4 Hz VTA power was absent in D2R-OE mice (Fig. 2d, e). An increase in 4 Hz power during the choice phase was also observed in the PFC in control animals but not in D2R-OE mice (Fig. 2f–h). These results suggest that a failure to recruit 4 Hz oscillations in the VTA and PFC may underlie their WM impairments.

**Impaired 4 Hz VTA-PFC synchrony during WM in D2R-OE mice.** Simultaneous LFP recordings in the VTA and PFC revealed that 4 Hz oscillations were often synchronized between the two structures (Fig. 3a). Notably, this VTA-PFC 4 Hz coherence increased selectively during the choice phase in control animals (Fig. 3b, d) but not in D2R-OE mice (Fig. 3c, d). Because 4 Hz

power was lower in D2R-OE mice in the choice phase it is possible that this could have resulted in less accurate phase estimates and thus lower coherence. As a complementary approach, we therefore quantified VTA-PFC coordination by calculating correlations between 4 Hz power in the two structures, which does not require phase to be estimated. This revealed that 4 Hz VTA-PFC power correlations increased in the choice phase in control but not D2R-OE animals (Supplementary Fig. 3). Taken together, these results demonstrate that not only the recruitment of 4 Hz oscillations during WM is impaired in D2R-OE mice, but also their coordination between the VTA and PFC.

D2R-OE mice did not display gross motor impairments, but their running speed during task performance was ~15% lower than in control animals (Supplementary Fig. 2a, b). However, we did not find a significant relationship between running speed and 4 Hz power or coherence in the choice phase (Supplementary Fig. 2c–e), changes in these variables between sample and choice phases (Supplementary Fig. 2f–h) or learning performance (Supplementary Fig. 2i). The impairments in behavior and neuronal recruitment in D2R-OE mice are therefore unlikely to be a consequence of motor impairments.

Interestingly, although D2R-OE mice failed to recruit 4 Hz VTA-PFC synchrony during WM, they nonetheless performed the task at the same level as control mice (Fig. 1d), albeit after more days of training (Fig. 1c). We therefore hypothesized that a

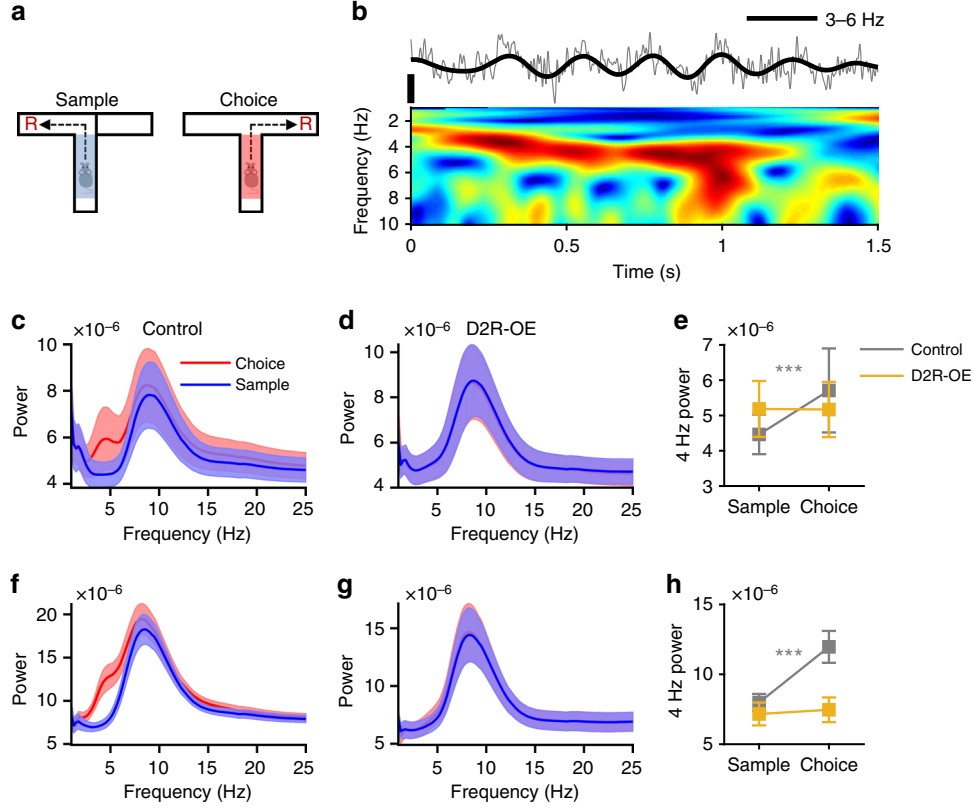

**Fig. 2** Impaired recruitment of 4 Hz oscillations in the VTA and PFC during working memory in D2R-OE mice. **a** Neural activity was compared in the center arm of the maze in the sample (blue) and choice phases (red). Overt behavior is the same in both phases but working memory is only required in the choice phase. **b** Example LFP trace recorded in the VTA (top) during the choice phase and its spectrogram (bottom). Note the presence of 3–6 Hz oscillations, referred to as "4 Hz" oscillations, which can also be observed in the filtered LFP trace (thick lines). Scale bar: 100 μV. **c, d** Average spectral power in the VTA during the sample and choice phases for control (**c**) and D2R-OE (**d**) mice. **e** Average strength of 4 Hz oscillations (calculated by averaging power between 3 and 6 Hz) during sample and choice phases in the two groups. Four Hertz oscillations were stronger during the choice phase in the VTA of control animals ($n = 19$, ***$p < .001$, sign-rank test) but not in D2R-OE mice ($n = 17$, $p = 0.79$, sign-rank test). **f–h** Four Hertz power in the PFC of control and D2R-OE mice during sample and choice phases, shown as in **c–e**. 4 Hz oscillations were stronger during the choice phase in the PFC of control animals ($n = 19$, ***$p < .001$, sign-rank test) but not in D2R-OE mice ($n = 17$, $p = 0.18$, sign-rank test). Shaded regions and error bars represent mean ± s.e.m over animals

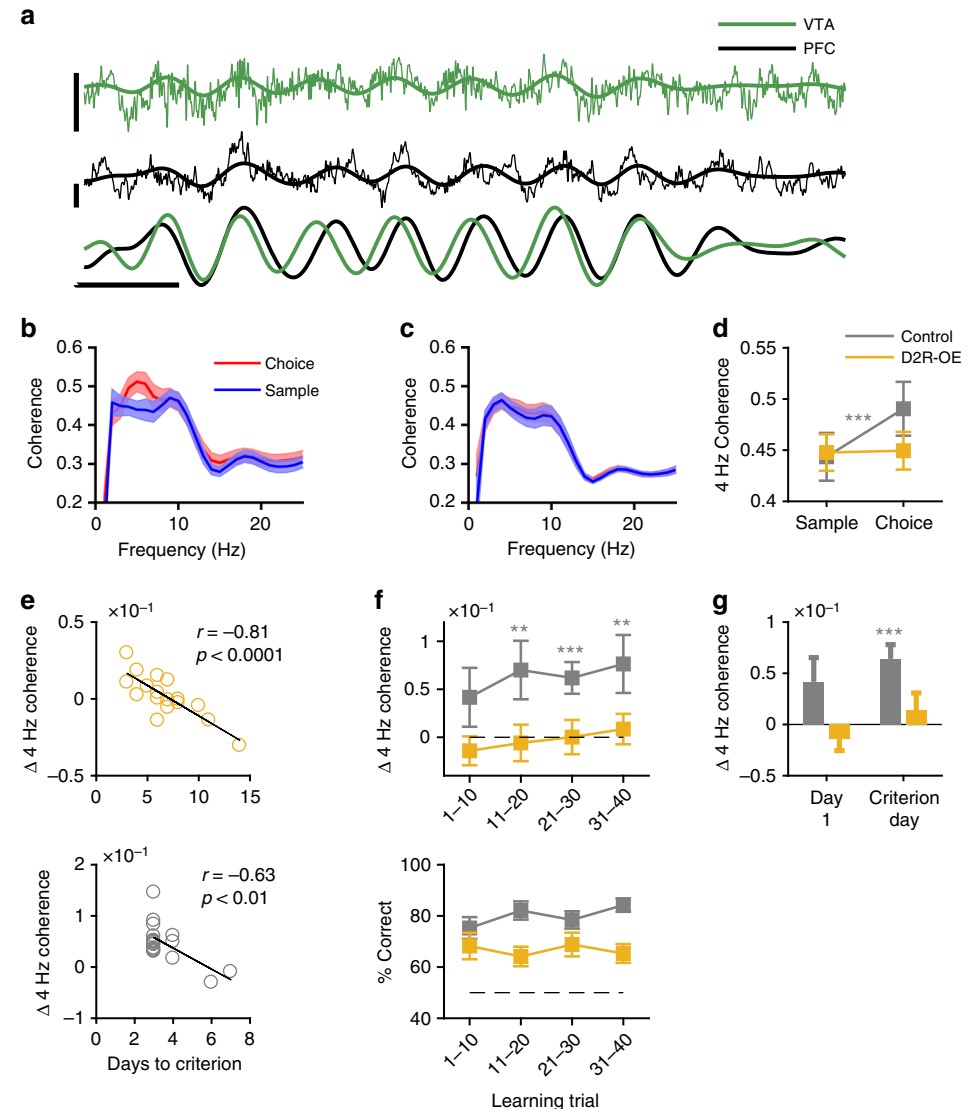

**Fig. 3** Working memory deficits are associated with impaired 4 Hz VTA-PFC synchrony in D2R-OE mice. **a** Example of simultaneously recorded LFP traces in the VTA (top, green) and PFC (middle, black). Thin lines represent the raw LFP whereas thick lines represent the LFP signal filtered between 3 and 6 Hz. The bottom graph shows the normalized 3–6 Hz filtered signals in the two structures, revealing coherent 4 Hz oscillations. Scale bar: 250 ms; 0.2 mV. **b**, **c** LFP coherence in sample and choice phases in control (**b**) and D2R-OE (**c**) mice. **d** Four Hertz coherence (calculated by averaging between 3 and 6 Hz) during sample and choice phases in the two genotypes. A selective increase in 4 Hz coherence was observed in control ($n = 19$, ***$p < .001$, sign-rank test) but not D2R-OE ($n = 17$, $p = 0.62$) animals. **e** Change in 4 Hz VTA-PFC coherence (choice minus sample, $\Delta$) plotted against learning rate (days to criterion) for each D2R-OE (top) and control (bottom) mouse. Animals that showed a greater increase in coherence in the choice phase learned the task more rapidly (D2R-OE: $r = -0.81$, $p < .0001$; Control: $r = -0.63$, $p < .01$, Pearson's correlation). **f** Change in coherence from sample to choice (top) and behavioral performance (bottom) during the first 40 trials of training (4 days). Control animals showed an increase in coherence only during trials 11–40 ($n = 19$, ***$p < .001$, **$p < .01$, sign-rank test). **g** Change in coherence on first day of training and the day when criterion performance was reached. D2R-OE mice did not show an increase in coherence during training, even on the day when criterion was reached ($n = 17$, $p = 0.62$). Shaded regions and error bars represent mean ± s.e.m measured across animals

failure to recruit and synchronize the VTA-PFC circuit might be responsible for the slower learning of the WM task in D2R-OE mice. Supporting this, we observed a strong correlation between learning rates and synchrony: D2R-OE mice that required the least number of sessions to learn the task also showed the greatest increase in VTA-PFC 4 Hz synchrony during the choice phase (Fig. 3e, top). A similar correlation was also observed in control animals (Fig. 3e, bottom), although they showed less variability in learning rates. Correlations were not observed between behavioral performance and changes in VTA-PFC coherence in other frequency bands (Supplementary Fig. 4). Taken together, these results point to a selective disruption of VTA-PFC 4 Hz

synchrony as a critical neural circuit deficit underlying WM impairments in D2R-OE mice.

The correlation between synchrony and learning suggests that a failure to recruit the VTA-PFC circuit may be present in D2R-OE mice during learning of the task. We therefore quantified VTA-PFC 4 Hz coherence in the same animals during the first 40 trials of learning (Fig. 3f; corresponding to training days 1–4; for animals that learned the task in 3 days, the first 10 test trials on day 4 were used for analysis). In control animals, VTA-PFC 4 Hz synchrony was similar in the sample and choice phases during the first 10 trials of training whereas on subsequent trials (11–40), coherence was larger in the choice phase (Fig. 3f). In contrast,

D2R-OE mice did not display any increase in VTA-PFC 4 Hz coherence during the learning trials or on the day when criterion performance was reached (Fig. 3g). These results suggest that the recruitment of VTA-PFC synchrony emerges during learning of the WM task in controls and that the failure of D2R-OE mice to recruit this synchrony could underlie their learning deficits.

**Impaired 4 Hz phase-locking of VTA DA neurons during WM in D2R-OE mice.** We next set out to examine how the deficit in VTA-PFC synchrony in D2R-OE mice is manifested at the cellular level. To this end, in a subset of animals we recorded the activity of VTA neurons (Control: $n = 387$ from 7 mice; D2R-OE: $n = 371$ from 7 mice), during performance of the WM task (Supplementary Fig. 1). The VTA contains different cell types, including DA neurons and gamma-aminobutyric acid (GABA) neurons[27]. We therefore separated VTA neurons into putative DA and GABA neurons using pharmacological and firing rate criteria similar to those used in previous studies[28] (see Methods, Supplementary Fig. 5). The number of classified putative VTA DA and GABA neurons was similar between the two genotypes (Supplementary Fig. 5). Consistent with our previous results in anesthetized animals[25], average firing rates of VTA DA neurons were slightly lower also in awake D2R-OE mice (recorded in the home cage; control: $3.62 \pm 0.24$ Hz, $n = 113$; D2R-OE: $3.03 \pm 0.20$ Hz, $n = 103$, $p < 0.01$, permutation test) as were the firing rates of GABA neurons (Control: $22.34 \pm 1.27$ Hz, $n = 124$; D2R-OE: $19.83 \pm 0.90$ Hz, $n = 104$, $p < 0.05$, permutation test).

The activity of individual neurons can be strongly modulated by the phase of ongoing LFP oscillations, a phenomenon known as "phase-locking"[29]. Indeed, during the WM task, the spiking of many VTA neurons was phase-locked to the 4 Hz oscillation, as manifest by their tendency to fire more at certain phases of the oscillation than others (Fig. 4a, b). We first quantified the phase-locking of each neuron using all of its recorded spikes (see Methods). This revealed that a large fraction of VTA DA and GABA neurons in control animals were significantly phase-locked to the 4 Hz oscillation recorded either in the VTA or PFC (Supplementary Fig. 6a–d). In D2R-OE mice, the proportion of significantly phase locked VTA DA and GABA neurons was comparable to that seen in controls and their preferred phase of firing was also similar (Supplementary Fig. 6a–d).

We next compared phase-locking strength in the center arm of the T-maze during the sample and choice phases. In control animals, the strength of phase-locking (see Methods) of VTA DA neurons to the VTA 4 Hz oscillations was higher in the choice phase whereas the phase locking strength of VTA GABA neurons did not differ between the two phases (Fig. 4c), suggesting that WM performance was associated with a DA cell-type specific increase in 4 Hz synchrony within the VTA. In contrast, in the goal arms of the T-maze—that is after animals had made their decision and WM was presumably no longer required—4 Hz phase-locking of DA neurons was not different between the sample and choice phase (Supplementary Fig. 7b). These results indicate that the increased entrainment of VTA DA neurons by 4 Hz oscillations in the choice phase is selectively associated with the task phase where WM-guided decision-making is required. In contrast to control animals, however, in D2R-OE mice the phase-locking strength of VTA DA neurons in the center arm was similar in sample and choice phases (Fig. 4c) whereas the phase locking of GABA neurons was actually lower in the choice phase (Fig. 4c).

Because of the deficit in VTA-PFC 4 Hz coherence (Fig. 3), we next examined the phase-locking of VTA neurons to PFC 4 Hz oscillations. In control animals, the phase-locking strength of VTA DA, but not GABA, neurons to PFC 4 Hz oscillations increased in the center arm during the choice phase (Fig. 4e). We also examined phase-locking to 4 Hz PFC oscillations in the goal arms and observed no difference between sample and choice phases for either VTA DA or GABA neurons (Supplementary Fig. 7e, f). Thus, 4 Hz oscillations support long-range synchrony selectively between VTA DA neurons and the PFC in a WM-dependent manner. Notably, this DA neuron-selective recruitment of long-range 4 Hz synchrony by WM was absent in D2R-OE mice (Fig. 4e). Taken together, these results demonstrate that the WM impairments in D2R-OE mice are associated with impaired WM-dependent entrainment of VTA DA neurons by 4 Hz oscillations within the VTA-PFC network.

**Impaired 4 Hz phase-locking of PFC neurons during WM in D2R-OE mice.** Because we also observed impaired recruitment of 4 Hz oscillations in the PFC of D2R-OE mice during WM (Fig. 2g), we examined whether 4 Hz phase-locking of PFC neurons might also be disrupted in these mice. To this end, we recorded the activity of single neurons in the PFC together with LFPs in the VTA in a subset of animals ($n = 190$ neurons from 8 D2R-OE mice and $n = 188$ neurons from 8 control mice). When all spikes from a recording session were taken into account, we found that many PFC neurons were phase-locked to the 4 Hz oscillation recorded in either the PFC or the VTA and that the number of phase-locked PFC neurons as well as their preferred phases were similar in the two genotypes (Supplementary Fig. 6e, f). Similar to VTA DA neurons, the phase-locking of PFC neurons to the 4 Hz rhythm in the PFC (Fig. 4d) and the VTA (Fig. 4f) increased during the choice phase in the center arm in control animals, further supporting the notion that the 4 Hz oscillation mediates neural synchrony between the two structures. Again, this increase in phase-locking was not observed in the goal arms of the T-maze (Supplementary Fig. 7d, g). Furthermore, the increase in 4 Hz phase-locking was absent in PFC neurons from D2R-OE mice (Fig. 4d, f), strengthening the conclusion that VTA-PFC synchrony deficits are manifest at the cellular level in these mice.

**Disrupted 4 Hz synchrony between VTA neurons in D2R-OE mice.** Oscillations in the LFP are generated by the coordinated activity patterns of neuronal populations[30,31]. However, the activity patterns underlying the VTA-PFC 4 Hz oscillation are not known. Strikingly, we found that many VTA neurons in control mice displayed clear rhythmic firing patterns when animals performed the WM task (Fig. 5a, b). To quantify these patterns we fitted the autocorrelogram of each neuron with a sinusoidal function and used the goodness of fit to determine whether a neuron displayed rhythmic firing and if so, at what frequency (see Methods). According to these criteria, many VTA neurons in control animals displayed oscillatory firing patterns and their oscillatory frequency was predominantly between 3 and 6 Hz (Median: 4.8 Hz; Fig. 5c), corresponding to the frequency range of the 4 Hz rhythm in the LFP (hereafter, this firing pattern will simply be referred to as "4 Hz firing"). In control animals, both DA and GABA neurons in the VTA displayed these 4 Hz firing patterns (Fig. 5c, d). Furthermore, rhythmic firing patterns were often coordinated between pairs of simultaneously recorded VTA neurons (Fig. 5a, b) and such rhythmic coordinated firing was also most frequently observed between 3 and 6 Hz (Fig. 5e, f). In contrast to VTA neurons, rhythmic 4 Hz firing patterns were largely absent in PFC neurons (Supplementary Fig. 8). Analysis of cross-correlations between 4 Hz LFP power in the VTA and PFC[32] furthermore revealed that PFC 4 Hz power was most strongly correlated with VTA 4 Hz power in the past (Supplementary Fig. 9), consistent with a directionality of influence from

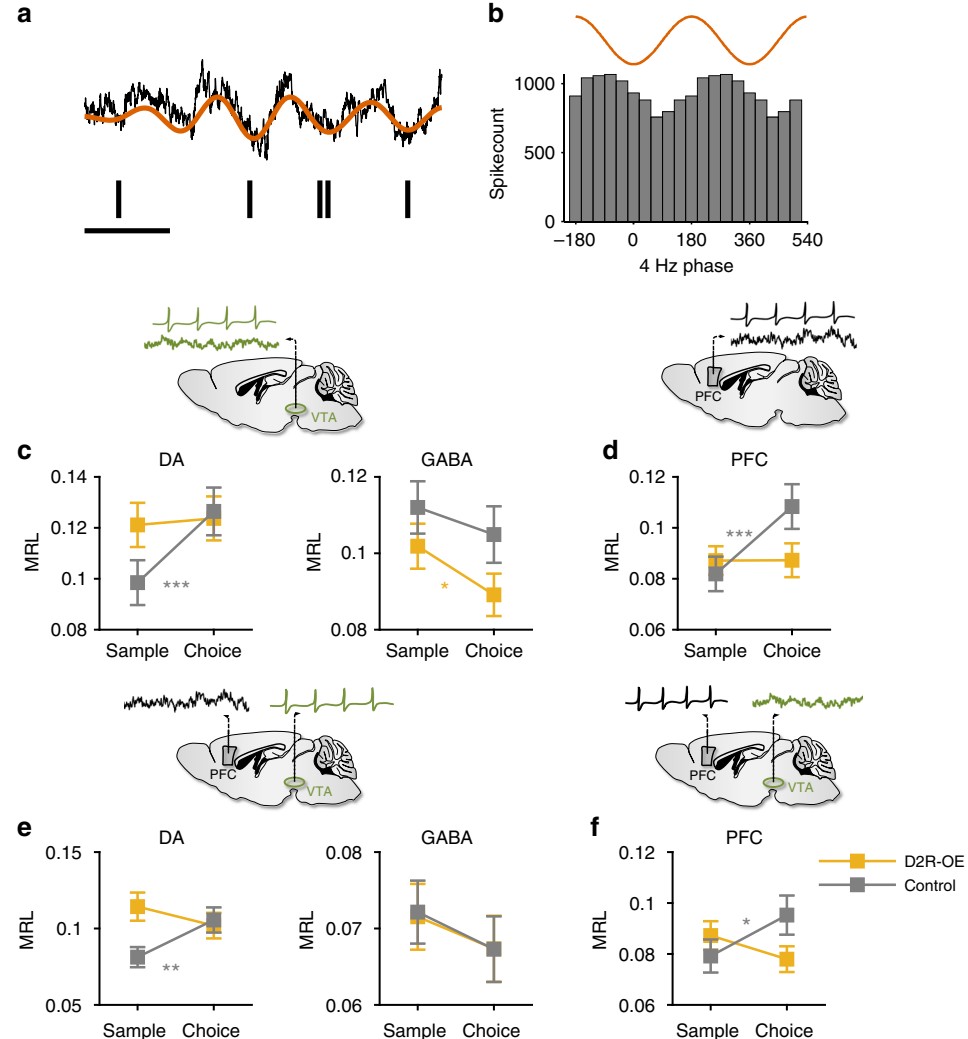

**Fig. 4** The recruitment of VTA DA and PFC neurons by 4 Hz oscillations during WM is disrupted in D2R-OE mice. **a** Example spike train of a VTA neuron (black lines) and the simultaneously recorded VTA LFP (top). The raw trace is shown in black and the same trace filtered between 3 and 6 Hz is shown in red. Scale bar: 250 ms. **b** Histogram of the phases of the 4 Hz oscillation at which the neuron in (**a**) spiked, indicating phase-locking to the trough of the 4 Hz oscillation. **c**, **d** Phase-locking of VTA DA (**c**, left, Control: $n = 75$; D2R-OE: $n = 68$), VTA GABA (**c**, right, Control: $n = 141$; D2R-OE: $n = 132$) and PFC (**d**, Control: $n = 74$; D2R-OE: $n = 98$) neurons to local 4 Hz oscillations in sample and choice phases. MRL: Mean resultant length (see Methods). Phase-locking of VTA DA and PFC neurons from control animals was stronger in the choice phase (***$p < .001$, sign-rank test) but not the phase-locking of VTA GABA neurons ($p = 0.13$). In D2R-OE mice an increase in phase-locking was not observed in VTA DA ($p = 0.83$) or PFC ($p = 0.67$) neurons whereas phase-locking was lower in VTA GABA neurons during the choice phase (*$p < .05$, sign-rank test). **e**, **f** Long-range phase-locking of VTA DA (**e**, left) and GABA (**e**, right) neurons to PFC 4 Hz oscillations and PFC neurons (**f**) to VTA 4 Hz oscillations. Long-range phase-locking was stronger in the choice phase in VTA DA and PFC neurons of control animals (**$p < .01$; *$p < .05$, sign-rank test) but not in VTA GABA neurons ($p = 0.13$). In D2R-OE mice an increase in long-range phase-locking was not observed (VTA DA: $p = 0.14$, VTA GABA: $p = 0.36$, PFC: $p = 0.12$). Error bars represent mean ± s.e.m measured across neurons. Brain diagrams in **c–f** are adapted with permission from ref.[62], Elsevier

the VTA to the PFC. These results suggest that 4 Hz firing patterns of VTA neurons may underlie the generation of 4 Hz LFP oscillations within the VTA-PFC network.

In D2R-OE mice the proportions of VTA DA and GABA neurons displaying rhythmic 4 Hz firing patterns were similar to those observed in control animals (Fig. 5c, d). In contrast, however, the coordination of 4 Hz firing between VTA neurons was selectively impaired (Fig. 5e, f): Fewer DA neuron pairs displayed coordinated 4 Hz activity compared to control mice, and the same was true for pairs of GABA neurons as well as DA-GABA neuron pairs (Fig. 5e, f). These results suggest that coordination of 4 Hz firing patterns among VTA neurons, but not the generation of these firing patterns per se, is disrupted in D2R-

OE mice. In turn, these deficits may contribute to the impairments in 4 Hz LFP oscillations within the VTA-PFC network in these mice.

## Selective disruption of task phase encoding in D2R-OE mice.
The firing rate of neurons during behavioral tasks is influenced by numerous events and variables, reflecting their involvement in the neural computations underlying task performance. We therefore speculated that a disruption in such task-related neural representations might contribute to the WM deficits in D2R-OE mice. To address this, we first examined the influence of WM demand by comparing neuronal firing rates in the center arm during sample and choice phases (Fig. 6a). In control animals, the

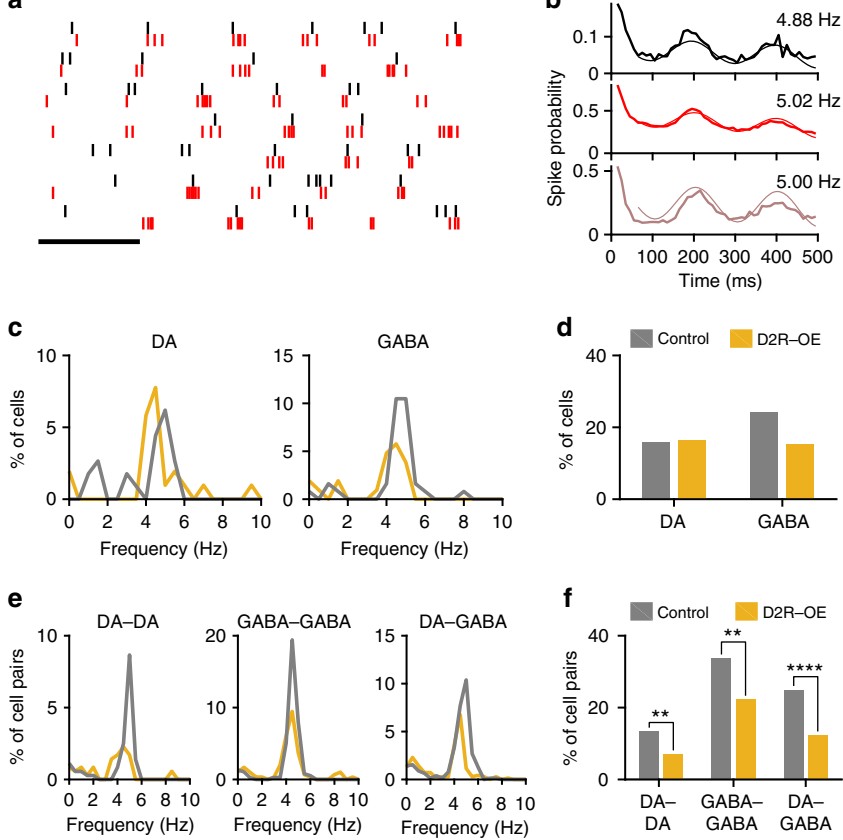

**Fig. 5** Disrupted coordination of rhythmic 4 Hz firing patterns in the VTA of D2R-OE mice. **a** Spikes of two simultaneously recorded VTA neurons, (red and black), during performance of the WM task. Each row corresponds to a single sample or choice run, aligned to run onset. Note the presence of coordinated rhythmic firing patterns in the two neurons. Scale bar: 250 ms. **b** Autocorrelograms of the two neurons in (**a**) (top, middle) and their cross-correlogram (bottom). Thin line indicates the sinusoidal function that best fit the correlograms and which was used to determine the presence of rhythmic firing and its frequency (inset; see Methods). **c** Distribution of frequencies at which rhythmic firing was observed in VTA DA and GABA neurons. Note the prevalence of firing patterns between 3 and 6 Hz, referred to here as "4 Hz" firing. **d** The percentage of VTA DA (Control: $n = 113$; D2R-OE: $n = 103$) and GABA neurons (Control: $n = 124$; D2R-OE: $n = 104$) displaying rhythmic firing in the 4 Hz range. The proportion of neurons was the same in both genotypes for both VTA DA ($p = 1.00$, Fisher's exact test) and GABA neurons ($p = 0.1$). **e** Distribution of frequencies at which VTA neuron pairs displayed rhythmic coordinated firing. **f** Proportion of DA-DA (Control: $n = 358$; D2R-OE: $n = 352$), GABA-GABA (Control: $n = 376$; D2R-OE: $n = 296$), and DA-GABA (Control: $n = 780$; D2R-OE: $n = 696$) cell pairs showing coordinated 4 Hz firing. The proportion of cell pairs showing rhythmic coordinated 4 Hz firing was lower in D2R-OE mice (**$p < .01$, ****$p < .0001$, Fisher's exact test)

activity of many VTA neurons was higher in either the choice phase or the sample phase (Fig. 6b, c). However, in D2R-OE mice the number of such phase-selective DA neurons was considerably lower (Fig. 6c) and did not differ significantly from chance levels ($p = .07$, binomial test). In contrast, the proportion of phase-selective GABA neurons was not affected in D2R-OE mice, indicating a cell-type specific impairment. We also found that many neurons in the PFC of control animals displayed task-phase specific responses and, similar to VTA DA neurons, that their proportion was significantly lower in D2R-OE mice (Fig. 6c). As a complementary approach, we trained a linear decoder to classify task phase based on the activity of all simultaneously recorded neurons (see Methods). Task phase could be decoded at above-chance levels in control animals, but decoding accuracy was significantly lower in D2R-OE mice based on the activity of VTA DA or PFC neurons but not when using the activity of VTA GABA neurons (Fig. 6d, e). Taken together, these results provide additional evidence for a failure to recruit VTA DA and PFC neurons during WM in D2R-OE mice.

Could the impairment in WM task phase representation reflect a more general disruption in task-related firing patterns? To

address this, we examined the influence of two other task variables on neuronal firing rates. First, we examined the influence of the animals' behavioral choice, that is whether they made a turn to the left or the right in the T-maze. In the VTA many neurons fired more strongly on either leftward trials or rightward trials, most prominently at the choice point of the maze (Fig. 7a, b). To quantify this, we compared firing rates on leftward and rightward trials around turn onset for each neuron (see Methods). This revealed turn-selective responses in a large fraction of VTA neurons in control animals and their proportion was similar in D2R-OE mice (Fig. 7c). The proportion of turn-selective PFC neurons was also similar in the two genotypes (Fig. 7c). Finally, we examined neural responses to rewards, which exert a strong influence over firing rates particularly in the VTA[14,33] (Fig. 7d). The firing rate of many VTA and PFC neurons increased significantly either shortly before or shortly after arrival at the reward zone (Fig. 7e; see Methods). Again, the proportion of reward-responsive neurons did not differ significantly between control and D2R-OE mice for either VTA or PFC neurons (Fig. 7f). Taken together, these results suggest that the representation of WM demand or task-phase, but not behavioral

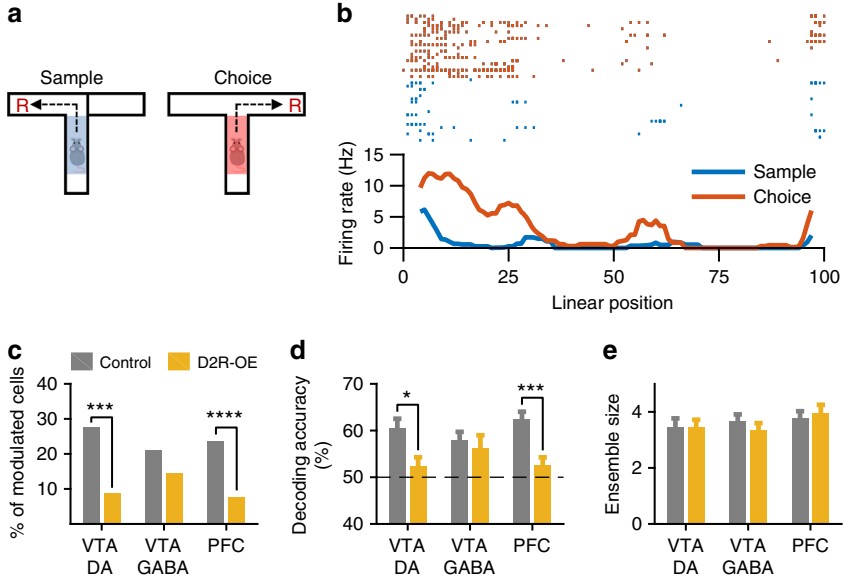

**Fig. 6** Selective disruption of task phase encoding in VTA DA and PFC neurons of D2R-OE mice. **a** Firing rates of VTA and PFC neurons were compared in the center arm of the maze between sample and choice phases. **b** Example neuron that was more active in the choice phase than in the sample phase (linear position 50 corresponds approximately to the decision point of the T-maze). **c** Proportion of VTA DA (Control: $n = 113$; D2R-OE: $n = 102$), VTA GABA (Control: $n = 124$; D2R-OE: $n = 104$) and PFC (Control: $n = 187$; D2R-OE: $n = 185$) neurons whose firing rate differed significantly in the center arm during sample and choice phases ($p < .05$, rank-sum test). The percentage of phase-selective VTA DA and PFC neurons was lower in D2R-OE mice (***$p < .001$, ****$p < .0001$, Fisher's exact test), but not the percentage of VTA GABA neurons ($p = 0.23$). **d** Accuracy (% correct) with which task phase (sample or choice) could be decoded from the activity of VTA and PFC neurons using a linear classifier (see Methods). Dashed line indicates chance level. Error bars represent mean ± s.e.m decoding accuracy over sessions (Control/D2R-OE: $n = 29/26$, 32/30 and 47/45 sessions for VTA DA, VTA GABA and PFC neurons, respectively). Decoding based on the activity of VTA DA and PFC neurons was significantly impaired in D2R-OE mice (*$p < .05$; ***$p < .001$, rank-sum test), but decoding based on VTA GABA neurons was not affected ($p = 0.99$). **e** Mean ± s.e.m number of neurons in each session that were used for decoding analysis in **d**. These numbers did not differ between groups ($p > 0.05$, rank-sum test)

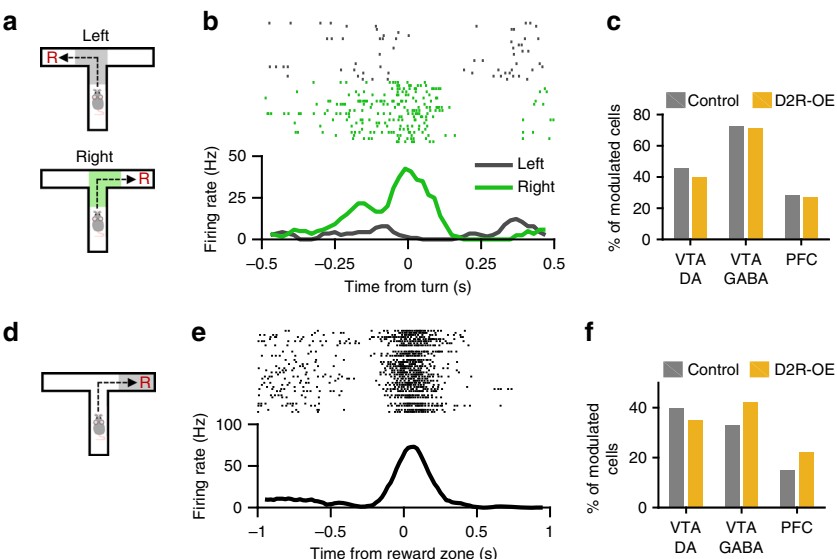

**Fig. 7** Choice-related and reward-related activity of VTA and PFC neurons in D2R-OE mice. **a** Firing rates were compared between leftward and rightward turns during the WM task. **b** Example of a VTA neuron that was more active when the mouse made a rightward turn. **c** The proportion of VTA DA (Control: $n = 113$; D2R-OE: $n = 102$), VTA GABA (Control: 124; D2R-OE: $n = 104$) and PFC (Control: $n = 187$; D2R-OE: $n = 185$) neurons whose firing rate differed significantly between left and right turns ($-500$ to $+500$ ms around turn onset; see Methods) did not differ between the two genotypes (VTA DA: $p = 0.41$, VTA GABA: $p = 0.88$, PFC: $p = 0.73$, Fisher's exact test). **d** Firing rates were examined around the time when animals reached the reward zone (R) in the T-maze. **e** Example of a VTA neuron responding to reward at the end of the goal arm. **f** Proportion of reward-responsive neurons ($n$ values as in **c**; see Methods) in VTA and PFC did not differ between the two genotypes (VTA DA: $p = 0.48$, VTA GABA: $p = 0.17$, PFC: $p = 0.08$, Fisher's exact test)

choice or reward, is selectively impaired in the VTA and PFC of D2R-OE mice. Finally, we examined the degree of overlap between the neuronal subpopulations belonging to these three response categories (Supplementary Fig. 10). Although some neurons were modulated by two or more task variables, their percentages were similar to what would be expected if these categories were independent. The majority of neurons were modulated exclusively by one of the three task variables, suggesting that separate subpopulations represent these three task variables and that one of these subpopulations (representing WM demand) is disrupted in D2R-OE mice.

## Discussion

Alterations in DA neurotransmission are widely viewed as contributing to cognitive dysfunction yet exactly how this manifests at the level of electrical activity of DA neurons has remained elusive. We found that the recruitment of VTA DA neurons that occurs on multiple levels during WM—manifesting in WM-selective activity patterns as well as in increased local synchrony and long-range coordination with the PFC—was absent in D2R-OE mice, which display WM deficits. These results thus define for the first time which disruptions in the electrophysiological activity of DA neurons might contribute to cognitive dysfunction.

In agreement with previous studies in rats[7], we observed that a 4 Hz LFP oscillation synchronizes activity between the VTA and PFC during WM performance in mice. This was observed both in 4 Hz LFP coherence as well as the phase-locking of VTA and PFC neurons to the 4 Hz oscillation recorded in both structures. In control animals, this 4 Hz synchrony was stronger during the choice phase of the task, when animals had to use WM to guide their behavior. In D2R-OE mice the percentage of neurons phase-locked to the 4 Hz oscillation (quantified using all the spikes in a session), as well as their preferred phase of firing, was similar to control animals, suggesting that the basic properties of 4 Hz VTA-PFC synchrony were not altered. Strikingly, however, the WM-dependent increase in VTA-PFC 4 Hz synchrony was absent in these mice. This suggests that the flexible recruitment of the VTA-PFC circuit by the WM demands of the task is selectively impaired in D2R-OE mice, which may ultimately contribute to their behavioral deficits.

As reported previously[23], D2R-OE mice took longer to learn the WM task than control animals. Although the synchrony deficits were observed during sessions in which D2R-OE mice had learned the task to the same level of performance as controls, they nonetheless correlated strongly with their learning deficits: D2R-OE mice that learned the WM task faster also showed a greater increase in 4 Hz synchrony during the choice phase of the task. A similar relationship was observed in control animals. Furthermore, during learning of the task the recruitment of VTA-PFC 4 Hz coherence by the WM demands of the task emerged in control animals but not in D2R-OE mice. These results suggest that synchrony within the VTA-PFC circuit supports the rapid learning of the WM task and that the slower learning in D2R-OE mice is caused by their failure to recruit this circuit during behavior. Nonetheless, D2R-OE mice are eventually able to learn this task, suggesting that their behavior may rely on a different circuit. Indeed, their slower learning is similar to what is seen in mice with PFC lesions[23]. One prediction would therefore be that D2R-OE mice will display WM-related neural activity (i.e., differential activity in sample and choice phases) in structures outside of the VTA-PFC circuit.

Although much progress has been made in understanding the cellular basis of LFP oscillations[34,35] the mechanisms underlying the 4 Hz VTA-PFC oscillation have remained elusive. Our results indicate that 4 Hz oscillations within the VTA-PFC network are generated by the rhythmic firing patterns of VTA neurons. During task engagement many DA and GABA neurons in the VTA fired rhythmically at 3–6 Hz, a frequency range similar to that of the 4 Hz LFP oscillation, and these firing patterns were often synchronized between VTA neurons. This suggests that rhythmic firing of VTA neurons generates 4 Hz oscillations in the VTA as well as in downstream targets including the PFC, as suggested by our cross-correlation analysis (Supplementary Fig. 9). Consistent with this possibility, blocking D1 DA receptors in the PFC reduces the amplitude of PFC 4 Hz oscillations[36]. VTA GABA neurons, some of which also project to the PFC[37], are likely to play a role as well.

In D2R-OE mice, VTA neurons displayed 4 Hz firing patterns to the same extent as control mice, suggesting that the mechanisms of 4 Hz rhythmogenesis are not affected per se. In contrast, the coordination of these firing patterns between neurons was selectively impaired. This suggests that the alterations in 4 Hz LFP oscillations in D2R-OE mice may be rooted in disrupted coordination among individual VTA neurons. How this coordination within the VTA is achieved mechanistically and how it is impaired in D2R-OE mice will be important questions for future research. In other brain regions, GABA neurons have been shown to synchronize neurons and thus generate LFP oscillations[34,35]. GABA neurons in the VTA, which exert a strong influence over the activity of DA neurons[27,38] and are coupled via gap junctions[39], could thus play an important role in generating 4 Hz oscillations.

In addition to disruptions in synchrony, our results also reveal selective deficits in task-related activity patterns in D2R-OE mice. Notably, many VTA DA and GABA neurons were differentially active during the sample and choice phases of our WM task, which could reflect the different WM demands in the two phases. Strikingly, the number of VTA DA and PFC neurons displaying phase-selective activity was greatly reduced in D2R-OE mice, whereas the number of phase-selective VTA GABA neurons was not affected. Taken together, these results provide strong evidence for the involvement of DA neuron activity in cognitive processing, as already suggested by primate studies[6], and also highlight their role in cognitive dysfunction. Importantly, neither the number of choice-selective or reward-responsive VTA and PFC neurons were altered in D2R-OE mice. The neural activity deficits in these mice are thus highly specific, both in terms of which task representations and neuronal subpopulations are affected. The affected DA neuron subpopulation might have a unique projection target[40], which will be important to identify in future studies. The selective deficits in task-related firing may reflect disruptions in specific inputs to these subpopulations, as suggested by studies in the PFC showing that different task-related activity patterns are mediated by specific afferent projections[41]. Interestingly, task-phase representations in the PFC are sensitive to DA levels[42] raising the possibility that the alterations in DA neurons might be responsible for the deficits in task-phase representation in PFC neurons of D2R-OE mice.

Alterations in DA signaling are central to the pathophysiology of several psychiatric disorders, including schizophrenia and depression[43,44]. In patients with schizophrenia, the density and occupancy of D2 DA receptors in the striatum is increased[24,45], although this may partly be a consequence of antipsychotic treatment[46]. The D2R-OE mouse line was generated to examine the pathophysiological consequences of elevated striatal D2 signaling and has been shown to display deficits in spatial WM[23], cognitive flexibility and interval timing[23,47–49]. Although cellular and molecular alterations in the PFC, VTA and striatum of D2R-OE mice have been documented in vitro and in vivo in anesthetized animals[25,26,50], our study is the first to demonstrate abnormalities in the recruitment of the VTA-PFC circuit during behavior and

adds to a growing body of literature demonstrating impairments in long-range synchrony and cognition in animal models of psychiatric disease[51]. Altered interactions between the PFC and the DA system have long been viewed as central to the pathophysiology of schizophrenia[52]. In schizophrenia patients, reduced functional connectivity has been observed between the PFC and the substantia nigra, another important source of DA that lies adjacent to the VTA[53]. Furthermore, in patients PFC activity correlates with DA release in the striatum[54,55]. Our results show how disrupted coupling between the DA system and the PFC can manifest itself at the cellular level.

A key outstanding question is how a striatal-specific increase in D2 receptors alters the function of the VTA and PFC. VTA neurons receive strong input from GABAergic medium spiny neurons (MSNs) in the striatum[56,57], whose intrinsic excitability is increased in D2R-OE mice[50]. VTA neurons in D2R-OE mice may thus receive stronger inhibitory input from the striatum, which could explain the reduction in firing rates we observed. In turn, these alterations in VTA neurons may affect their downstream targets, including the PFC. Our analysis suggests that 4 Hz oscillations within the VTA-PFC network—which play a key role in synchronizing activity during WM—are likely generated within the VTA and transmitted to the PFC. Alterations in the anatomical projections between the VTA and PFC[8,58,59] in D2R-OE mice may also play a role in the reduced synchrony between them, as has been shown for example for hippocampal-prefrontal synchrony deficits in animal models of psychiatric illness[22,60].

## Methods

**Animals**. All procedures were conducted in accordance with the guidelines of the German Animal Protection Act and were approved by the local authorities (Regierungspräsidium Darmstadt). D2R-OE mice were generated as described previously[23] and bred at the New York State Psychiatric Institute in accordance with the NYSPI IACUC. Briefly, mice expressing the human D2R under control of the tetO promoter were generated on a C57BL/6-CBA(J) background and back-crossed for >10 generations to the C57BL/6(J) background. Crossing these animals with mice expressing the tetracycline transactivator (tTA) transgene under the calcium/calmodulin-dependent kinase IIα (CaMKIIα) promoter on a 129SveV Tac background (backcrossed >20 generations) resulted in double-transgenic mice expressing human D2Rs selectively in striatal medium spiny neurons. Littermates carrying a single transgene (tetO-hD2R or CaMKIIα-tTA) or no transgene were used as controls. Male adult D2R-OE mice (3–8 months at the start of experiments) paired with their control littermates were used. Animals were kept in a ventilated animal scantainer on a 12-h light/dark cycle. All experiments were performed during the light cycle.

**Surgical procedures**. Animals were anesthetized using isoflurane (1–2%) and placed in a stereotaxic frame. At the onset of anesthesia, all animals received subcutaneous injections of carprofen (4 mg/kg) and dexamethasone (2 mg/kg) and an intraperitoneal injection of atropine (0.1 mg/kg). The animals' temperature was maintained for the duration of the surgical procedure using a heating blanket. Anesthesia levels were monitored throughout the surgery and the concentration of isoflurane adjusted so that the breathing rate never fell below 1 Hz. After exposing the skull surface, craniotomies were made overlying the medial PFC (1.95 mm anterior to bregma, 0.4 mm lateral to the midline) and the VTA (3.3 mm posterior to bregma, 0.45 mm lateral to the midline). For recording the activity of single neurons we used a moveable bundle of 5–6 stereotrodes made by twisting together two 0.0005 inch tungsten wires (M338350, California Fine Wire). The stereotrode bundle was attached to a custom-made microdrive that made it possible to advance the electrodes along the dorsoventral axis. In 14 animals (7 controls, 7 D2R-OEs), the stereotrode bundle was inserted through the craniotomy above the VTA to a depth of 4 mm below bregma and a single .003 inch tungsten wire (M279540, California Fine Wire) for recording LFPs was inserted through the craniotomy above the PFC to a depth of 2.3 mm below bregma. In 16 animals (8 controls, 8 D2R-OEs) the stereotrode array was implanted into the PFC to a depth of 1.6 mm below bregma and the single tungsten wire in the VTA to a depth of 4.4 below bregma. In 6 animals (4 controls, 2 D2R-OEs) LFP wires were implanted into both the VTA and PFC using the abovementioned coordinates. All electrode wires were connected to an electrode interface board (EIB-16; Neuralynx) for relaying electrophysiological signals to the data acquisition system. Skull screws over the cerebellum and frontal cortex served as reference and ground, respectively, and provided additional anchoring support for the microdrives and electrodes. Animals were individually housed and allowed to recover for 1 week following the operation.

**Behavioral task**. Following recovery from surgery, animals' food intake was restricted until their body weight reached 85% of their pre-surgical weight. Animals were then trained on a discrete non-match-to-sample T-maze test of spatial working memory (see Fig. 1a) as described previously[21]. The T-maze consisted of three arms (one center arm and two goal arms), each of which was 40 cm long, 4.5 cm wide, and with 4 cm high walls. Each trial of the task consisted of two phases. In the "sample phase", mice ran up the center arm of the maze and entered one of the goal arms to collect a reward (20 mg food pellet, F0071, Bio-Serv), while the other arm was blocked. After a brief delay (5–10 s), the mice again ran up the center arm, but now both goal arms were open ("choice phase"). To obtain a reward, animals had to enter the goal arm not visited during the sample phase. After 2 days of habituation to the maze and 2 days of shaping, animals were given daily training sessions of 10 trials until they reached criterion performance, defined as at least seven trials correct per day for three consecutive days. After this criterion had been reached, animals were given daily sessions of 20–25 trials. Although neural data were recorded throughout the experiment, the results presented here are from sessions after criterion was reached, with the exception of Fig. 3f, g. Stereotrodes in the VTA and PFC were advanced at least 40–80 μm daily to ensure that different populations of cells were recorded in each behavioral session.

**Electrophysiological recording**. Neural data (putative spikes and local field potentials) were acquired using a 16-channel headstage that was connected to the electrode interface board and which relayed the signals to the data acquisition system (Digital Lynx S, Neuralynx) running Cheetah acquisition software (Neuralynx). The animal's x and y position in the T-maze was also recorded using a light-emitting diode mounted on the head stage and digitized at 25 Hz using the same clock as for the electrophysiological data (Cheetah, Neuralynx). To extract putative spikes, neural signals were bandpass filtered between 0.6 and 6 kHz, and waveforms that passed a threshold were digitized at 30 kHz. Waveforms were then sorted offline into single-unit clusters using SPIKESORT3D (Neuralynx). To extract local field potential (LFP) activity, the same signals were bandpass filtered between 1 and 1000 Hz and digitized at 2 kHz. Subsequent analysis of neural and behavioral data was performed using scripts custom-written in MATLAB (MathWorks).

**Analysis of behavioral data**. During each sample and choice phase, animals ran from the bottom of the center arm to the end of one of the goal arms. Prior to subsequent analysis we first converted the two-dimensional trajectory of each sample and choice run to a one-dimensional linear trajectory. This linearized trajectory was represented in units of normalized position where position 1 corresponded to the bottom of the center arm, positions 50–60 corresponded approximately to the junction of the T-maze and position 100 to the end of the goal arm. Each sample of the animal's behavioral position (recorded at 25 Hz) during sample and choice runs was thus assigned a linear position value between 1 and 100. These linear positions were used for restricting subsequent analysis of neural data to specific positions in the maze. For most analysis, we examined activity in the center arm (linear positions 1–50) and in some cases in the goal arms (linear positions 51–100).

**Spectral analysis of LFPs**. The spectral power and coherence of LFPs recorded in the VTA and PFC was calculated using the continuous wavelet transform. Briefly, the LFP was convolved with a series of Morlet wavelets with center frequencies between 1 and 25 Hz and a length of three cycles. The result of these convolutions is the wavelet transform of the LFP, WTLFP(f,t), a matrix of complex numbers (or vectors) whose absolute values (or length) and arguments (or angles) represent the amplitude and phase, respectively, of the LFP at frequency f and time t. The spectrogram of the LFP, representing spectral power as a function of frequency and time was thus defined as the absolute value of WTLFP(f,t). To compute coherence we first computed WTLFP(f,t) for the VTA and PFC and then calculated the cross-spectrum of the two wavelet transforms. At each frequency f and time t, coherence was then estimated by calculating the sum of the cross-spectrum at six surrounding timepoints spaced one cycle apart, taking the absolute value of the sum and dividing it by the square root of the products of the autospectra of the two signals. The coherence thus measures how consistent the phase-relationship is between the LFPs in the two structures over a period of 6 cycles for each frequency f and for each timepoint t. We also measured the cross-correlations between 4 Hz power fluctuations in the LFPs from the VTA and PFC (Supplementary Fig. 9). This was done by averaging the spectrograms for VTA and PFC between 3 and 6 Hz, yielding a time series of 4 Hz power over time in the two structures. The correlation coefficient of these two time series was then computed at different time lags by shifting one signal relative to the other (±250 ms in 5 ms steps). This analysis can determine whether changes in power in one structure lead or lag changes in another structure and thus reveal the directionality of their interactions[32].

In order to calculate power and coherence as a function of the animal's position in the T-maze we first determined for each behavioral sample the LFP sample closest to it in time. The linear position corresponding to each behavioral sample was then assigned the power and coherence values of the closest LFP sample. For each linear position, power and coherence were then averaged, separately for sample and choice phases, over all trials from each animal. For statistical analysis, power and coherence were averaged within the frequency band of the 4 Hz

oscillation (3–6 Hz; referred to as "4 Hz power" and "4 Hz coherence") as well as within the part of the maze corresponding to the center arm (linear positions 1–50) and compared between sample and choice phases.

**Classification of VTA neurons**. Neurons recorded from the VTA were classified as either putative DA or GABA neurons based on their baseline firing rates and their response to the D2 receptor agonist apomorphine[28]. Immediately following each recording during the WM task animals were placed in their home cage and baseline neural activity recorded for approximately 10 min. Animals were then injected with 1–1.5 mg/kg apomorphine (i.p.) and neural activity recorded for a further 15 min. Neurons were classified as putative DA neurons if they met two criteria: (1) their baseline firing rate prior to apomorphine administration was <10 Hz and (2) apomorphine injection caused a 50% or greater decrease in firing rate. Neurons whose baseline firing rate was above 10 Hz were classified as putative GABA neurons. Neurons whose baseline rate was <10 Hz but whose firing rate was either enhanced, suppressed by less than 50% or unchanged following apomorphine administration were labeled as "unclassified" and not included in further analysis.

**Phase-locking analysis**. LFPs were filtered in the 4 Hz range (3–6 Hz) using a finite impulse response filter (fir1; Matlab, Mathworks) and the filtered signal was shifted to compensate for the phase-delay of the filter. The phase and amplitude of each LFP sample was then computed using the Hilbert transform (hilbert; Matlab), and each spike was assigned the phase of the LFP sample closest to it in time. Phase locking was quantified as the circular concentration of the phase distribution of all spikes, which we defined as the mean resultant length (MRL) of the phase angles. MRL is the sum of unit length vectors representing the phases at which each spike occurred divided by the number of spikes. Therefore, it takes on values between 0 (no phase locking) and 1 (perfect phase locking). The statistical significance of phase locking was assessed using Rayleigh's test for circular uniformity. Sample estimates of MRL suffer from a positive bias (that is, they overestimate the population MRL) whose magnitude is highly dependent on sample size. When comparing phase locking in the sample and choice phases of the task (see Fig. 4), we therefore adopted two measures to limit the effects of this bias on our analysis. First, we only computed phase locking for cells firing at least 50 spikes in both task phases. Second, we subsampled spikes from the task phase with more spikes so that equal numbers of spikes were used for calculating MRL in the two phases. Sub-samples were drawn 1000 times for each neuron and MRL values averaged across subsamples.

**Analysis of rhythmic firing patterns**. To analyze whether spike trains in the VTA and PFC displayed rhythmic firing, we first computed the autocorrelation of each neuron 10–500 ms after each spike in 10 ms bins. We then fit each autocorrelogram with the sum of an exponential and a sinusoidal function:

$$a * \exp(-b * t^2) + c * \sin(d * t + e),$$

where $t$ is time. The autocorrelograms were fit using nonlinear least-squares regression in Matlab (Mathworks). To quantify the degree to which a neuron displayed rhythmic firing we calculated the circular correlation $r$ between its autocorrelogram and the sinusoidal component of the fit (function circ_corrcl in the CircStat toolbox[61]). This $r$ value was then compared against the $r$ values of surrogate spike trains that were generated by jittering the spike timestamps of the neuron between −1000 and +1000 ms. For each neuron we generated 1000 surrogate spike trains and calculated their autocorrelation and best-fit $r$ values in the same way as for the real data. A neuron was classified as showing significant rhythmic firing if its $r$ value was greater than 95% of the $r$ values of its surrogate spike trains ($p < .05$). The oscillation frequency of the neuron was then determined from the frequency of the sinusoidal component of the fit (determined by parameter $d$ in the above equation). We used a similar approach to ask whether oscillatory firing was correlated between pairs of neurons. First, we computed the cross correlogram of each pair of simultaneously recorded neurons 500 ms before and after each spike in 10 ms bins. Cross correlograms were then fitted with the same function described above for autocorrelograms and their circular correlation $r$ with the sinusoidal component of the fit computed. Cell pairs were classified as displaying rhythmic co-firing if the strength of their $r$ value exceeded 95% of the $r$ values obtained from their surrogate spike trains, which were jittered in the same way as described above for autocorrelations. The frequency of rhythmic co-firing was furthermore determined from the sinusoidal fit. Rhythmic firing patterns of individual neurons and neuron pairs were classified as "4 Hz" if their frequency was between 3 and 6 Hz.

**Analysis of task-related firing patterns**. For analyzing differences in firing rates between sample and choice phases (Fig. 6a), we calculated for each neuron its firing rate in the sample and choice phase of each trial, from the start of each run until the animal reached the decision point in the maze. We then classified neurons as phase-selective if their firing rates in the two phases differed significantly ($p < 0.05$, rank-sum test). As a complementary approach to examining task phase-related neural representation we trained a linear classifier to decode the identity of each task phase from the activity of VTA and PFC neurons. To this end, a matrix was constructed for each session representing the firing rates of each neuron in the sample and choice phases of each trial. A linear decoder (classify, Matlab) was then used to classify individual task phases as either sample or choice using leave-one-out cross-validation: for each individual task phase $p$, the classifier was trained using all task phases except $p$ and then tested on $p$. This was repeated for all task phases and decoding accuracy was quantified as the % of task phases that were correctly classified by the decoder. Decoding was performed separately for VTA DA, VTA GABA, and PFC neurons. For each neuron type, sessions were only included for analysis if at least 2 neurons of that type were recorded during the session.

For analyzing choice-related activity (Fig. 7) we examined the firing rate of each neuron around the time when the animals started making a leftward or a rightward turn at the T-maze junction. To determine this time point we first calculated the difference in the animal's horizontal position from one behavioral sample to the next. We then searched for the first 15 consecutive behavioral samples in which the animals moved consistently leftward or rightward; the timestamp of the first behavioral sample in this series was defined as the onset of the turn. This was done separately for each sample and choice run. We then computed the firing rate of each neuron from 500 ms before to 500 ms after the turning timepoint in 100 ms bins and compared leftward vs. rightward turns (sample and choice phases combined) with a 2-way ANOVA (Time X Direction). Neurons were classified as turn-selective if they showed a significant ($p < .05$) main effect of Direction or a Time X Direction interaction. For analyzing responses of VTA and PFC neurons to reward (Fig. 7), we first determined when animals arrived at the reward well located at the end of the goal arms, where reward could be retrieved in the sample and choice phases. We then quantified each neuron's response to reward by averaging its firing rate between −500 and +500 ms relative to arrival at the reward reward well. This time window was chosen because many neurons increased their firing rate shortly before arrival at the reward well, as has previously been described[14]. Neurons were considered reward-responsive if their firing rate during this time window was significantly higher than their firing rate 500–1000 ms before arrival at the reward zone ($p < .05$, sign-rank test).

**Histology**. To verify recording locations, current (50 mA, 10 s) was passed through the electrodes at the end of the experiment. Animals were then transcardially perfused with 4% paraformaldehyde and 15% picric acid in PBS. Brains were post-fixed overnight and coronal brain slices (60 µm) were sectioned using a vibratome (VT1000S, Leica). To localize the VTA, slices were immunostained for thyrosine hydroxylase (TH). Briefly, sections were rinsed with PBS and then incubated in a blocking solution (10% horse serum, 0.5% Triton X-100 and 0.2% BSA in PBS) for 2 h at room temperature. Slices were then incubated in a carrier solution (1% horse serum, 0.5% Triton X-100 and 0.2% BSA in PBS) containing the primary antibody (polyclonal rabbit anti-TH,1:1000; Calbiochem, Merck) overnight at room temperature. Following incubation in secondary antibody (AlexaFluor-488 goat anti-rabbit, 1:750; Life Technologies) overnight at room temperature, sections were washed with PBS, mounted on slides and coverslipped. Animals with incorrect electrode placement were excluded from analysis.

**Statistics**. Statistical analysis was done using MATLAB (Mathworks). Group data was represented as the mean ± standard error of the mean (s.e.m.). We did not statistically compare the variances between the groups. All statistical tests were two-tailed and had an α level of 0.05. We did not make any assumptions about normality and therefore used non-parametric tests. No statistical methods were used to predetermine sample size, but our sample sizes were comparable to or larger than those typically used in similar studies. The experimental groups in the study consist of D2R-OE transgenic mice and their control littermates. We therefore could not randomly assign animals to the experimental groups. The experimenter was not blinded to the group allocation.

**Code availability**. All relevant code supporting the findings of this study is available from the corresponding authors upon reasonable request

**Data availability**. All relevant data supporting the findings of this study are available from the corresponding authors upon reasonable request.

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

## Acknowledgements

We thank Beatrice Fisher, Jasmine Sonntag and Thomas Wulf for technical assistance. This work was supported by Priority Program SPP 1665 of the German Research Foundation (DU1433/1-1 to S.D. and J.R., SCHN1370/2-1 to G.S., SI 1942/1-1 to T.S.), an EMBO Long-term fellowship (ALTF_210-2012 to S.D.) and the Lieber Institute for Brain Development (to E.H.S. and E.R.K.).

## Author contributions

S.D., E.H.S., J.R. and T.S. designed the experiments. E.H.S. and E.R.K. generated and supplied the D2R-OE and control mice. S.D. performed all experiments. S.D. and T.S.

analyzed the data. G.S. contributed advice on analysis. S.D. and T.S. wrote the paper with comments from all of the authors.

## Additional information

**Competing interests:** The authors declare no competing interests.

