## [Peer Review File · Nature Communications]

Reviewers' comments:

Reviewer #1 (Remarks to the Author):

The manuscript by Duvarci et al. describes patterns of synchronization between the VTA and PFC during a T-maze working memory task, and associated abnormalities in transgenic mice which have impaired acquisition of this task. The authors find that the power and synchrony of 4 Hz oscillations in the VTA and PFC normally increases during the choice phase of the task, but that this does not occur in D2R-OE mice. The number of trials needed to learn the task also correlates with the 4 Hz VTA-PFC coherence. Phase locking of VTA DA neurons and PFC neurons to 4 Hz oscillations recorded in either structure normally increases during the choice phase, but this was also not observed in D2R-OE mice. VTA neurons also exhibit synchronization at 4 Hz, and the number of cell pairs with significant synchronization was also reduced in D2R-OE mice. Finally, the number of VTA DA neurons or PFC neurons which encode the task phase were both significantly reduced in D2R-OE mice. These findings show that 4 Hz-synchronization within the VTA and between the VTA and PFC is modulated by the T-maze task and abnormal in D2R-OE mice.

The results are interesting, and the experiments appear to be generally well done, but there are some issues with the analysis and presentation and that should be addressed. In particular, while these results nicely describe an abnormality, the exact significance of this abnormality for behavior is somewhat unclear (see below). I.e., it's not certainly not clear that these deficits cause the behavioral abnormalities, and it's not even clear how directly correlated they are since they are observed after mice complete learning, at which point the mutant mice no longer exhibit behavioral deficits. While I really do like this work, and think this is an exciting start / important new lead, I think these issues would need to be sorted out at least a bit before the work was published in a general interest journal such as Nature Communications. Otherwise, if the authors want to publish the paper without doing additional experiments to address these issues, I think a more specialized journal (e.g., a disease or neuroscience focused journal) would be appropriate.

1. The authors conclude that "altered DA neuron activity underlies cognitive impairments" and "impaired 4 Hz VTA-PFC synchrony underlies working memory deficits in D2R-OE mice." These conclusions are simply too strong. We do not know whether these abnormalities cause impaired learning in this task, whether they are reflective of impaired learning, or whether they are simply additional consequences of the underlying pathology. Simply observing a correlation between the number of days required for learning and the coherence does not establish a causal relationship – the penetrance of the genetic perturbation may simply vary across animals.
2. Coherence and phase locking both tend to decrease whenever power decreases, simply because it becomes more difficult to estimate the phase of the oscillation. Thus, it is very difficult to interpret the decreases in coherence and phase locking observed here. If the authors could show that correlated changes in the amplitude of the 4 Hz oscillation normally increase from the sample to choice phase, but that this does not occur in D2R-OE mice, this would be better as it should not be subject to the same confound.
3. I'm not sure exactly what to make of the method for estimating number of VTA neurons or VTA neuronal pairs that exhibit significant rhythmicity. Do the authors fit the combined Gaussian / sinusoid to the autocorrelation / cross-correlation, then use the frequency from this fit to construct a sinusoid, and finally test whether that sinusoid is correlated with the auto / cross-correlation? If this is what the authors did, then the p-value should be obtained by bootstrapping, i.e., comparison to appropriately shuffled datasets. Otherwise, I am concerned that the authors may be finding some spurious correlations just because they perform an optimization step (fitting parameters) before testing the

correlation. This will automatically induce some degree of correlation between the sinusoid and auto/cross-correlation.

4. Overall, it's not clear how the abnormalities in 4 Hz oscillations / synchrony relate to behavioral deficits. In particular, phase-locking of DA neurons to 4 Hz oscillations in either the VTA or PFC is not reduced in D2R-OE mice, it's simply no longer modulated by task phase (sample vs. choice). Rather than being glossed over, this point should be discussed.

5. The authors made a decision to examine the data only after animals had reached criterion for learning. I understand they did this because they want to explore activity in the context of similar behavioral performance. However, this choice makes it hard to determine the relationship of this activity to the behavioral deficits, since the data was not collected in the context of the behavioral deficits. For this reason, it would have been better to analyze data both during and following learning. For example, if the authors found that deficits in rhythmicity / synchrony were not present initially, but developed towards the end of training, then the conclusions / interpretation would change dramatically. Similar, if some of the other properties, e.g., the encoding of L vs. R by PFC and VTA neurons was initially abnormal but then normalized over the course of training, this too would dramatically alter the conclusions.

Reviewer #2 (Remarks to the Author):

Duvarci and colleagues reported rhythmic synchronous activity in PFC and midbrain VTA during a working memory task in wildtype and disease-relevant transgenic (D2R-OE) mice. They found increase of powers and coherences of 4-Hz oscillation in the PFC and VTA during working memory in wildtype mice but not in D2R-OE mice. The spiking activities of neurons in PFC and VTA were phase-locked to 4-Hz oscillation during the task. Importantly, task-phase-relevant neurons were more strongly modulated by 4-Hz oscillation in wildtype mice, but this was not observed in D2R-OE mice. These results are interesting and timely, and provides new insight into the functional relevance of 4-Hz oscillation in the PFC and VTA network. This study will make a significant contribution to the field, though several issues need to be addressed before publication.

1) One concern is about the relationship between 4-Hz oscillation and behavioral performance. The authors found that the D2R-OE mice had impairment on learning rate (Fig 1b, c) but they performed the task well after they learned it (Fig 1d). This result indicate that 4-Hz oscillation would reflect learning performance. However, recruitment and synchrony of 4-Hz oscillation in the PFC and VTA were only assessed after the mice learned the task (Fig 2 & 3). Since they did recording also in the learning stages (P. 24, Methods), additional analysis of LFP in those periods would more strongly support the link between 4-Hz and behavior.

2) About the result of the correlations between 4-Hz coherence and learning rate in Figure 3e. Here, the authors only showed the result of D2R-OE mice. Is it possible to include the data of control mice?

3) To rule out the possibility that the differences of neuronal activities between control and D2R-OE mice were due to motor impairments, quantification of running speeds of mice (or any motor related parameters) would be needed.

4) Related above, in the same animals, were the running speeds same between sample and test phases?

5) Please provide the explanations of the dotted lines in Fig 6c, d and Figure 7c, f.

Reviewer #1 (Remarks to the Author):

The manuscript by Duvarci et al. describes patterns of synchronization between the VTA and PFC during a T-maze working memory task, and associated abnormalities in transgenic mice which have impaired acquisition of this task. The authors find that the power and synchrony of 4 Hz oscillations in the VTA and PFC normally increases during the choice phase of the task, but that this does not occur in D2R-OE mice. The number of trials needed to learn the task also correlates with the 4 Hz VTA-PFC coherence. Phase locking of VTA DA neurons and PFC neurons to 4 Hz oscillations recorded in either structure normally increases during the choice phase, but this was also not observed in D2R-OE mice. VTA neurons also exhibit synchronization at 4 Hz, and the number of cell pairs with significant synchronization was also reduced in D2R-OE mice. Finally, the number of VTA DA neurons or PFC neurons which encode the task phase were both significantly reduced in D2R-OE mice. These findings show that 4 Hz-synchronization within the VTA and between the VTA and PFC is modulated by the T-maze task and abnormal in D2R-OE mice.

The results are interesting, and the experiments appear to be generally well done, but there are some issues with the analysis and presentation and that should be addressed. In particular, while these results nicely describe an abnormality, the exact significance of this abnormality for behavior is somewhat unclear (see below). I.e., it's not certainly not clear that these deficits cause the behavioral abnormalities, and it's not even clear how directly correlated they are since they are observed after mice complete learning, at which point the mutant mice no longer exhibit behavioral deficits. While I really do like this work, and think this is an exciting start / important new lead, I think these issues would need to be sorted out at least a bit before the work was published in a general interest journal such as Nature Communications. Otherwise, if the authors want to publish the paper without doing additional experiments to address these issues, I think a more specialized journal (e.g., a disease or neuroscience focused journal) would be appropriate.

1. The authors conclude that “altered DA neuron activity underlies cognitive impairments” and “impaired 4 Hz VTA-PFC synchrony underlies working memory deficits in D2R-OE mice.” These conclusions are simply too strong. We do not know whether these abnormalities cause impaired learning in this task, whether they are reflective of impaired learning, or whether they are simply additional consequences of the underlying pathology. Simply observing a correlation between the number of days required for learning and the coherence does not establish a causal relationship – the penetrance of the genetic perturbation may simply vary across animals.

Author's Response: We are grateful for the referee's very careful and detailed review of our study. The reviewer is correct that our results do not prove that there is a causal relationship between the synchrony deficits and the behavioral deficits. We agree that our conclusions were

too strong in certain places in the manuscript and have now modified them accordingly. Instead of “Impaired 4 Hz VTA-PFC synchrony underlies working memory deficits in D2R-OE mice” we now write “Impaired 4 Hz VTA-PFC synchrony is associated with working memory deficits in D2R-OE mice”. Instead of “underlies cognitive impairments” we now write “may underlie cognitive impairments”. We have also made sure that similarly strong statements were not made elsewhere in the manuscript.

Although our study does not establish a causal relationship between synchrony deficits and behavioral deficits, it nonetheless provides several lines of evidence that strongly suggest the two are related: 1) VTA-PFC synchrony increases during task phases requiring working memory in control mice but not in D2R-OE mice; 2) VTA-PFC synchrony increases correlate with learning performance in D2R-OE mice as well as in controls (new results included in response to reviewer #2, comment 2); 3) VTA-PFC synchrony increases emerge during learning in controls but not D2ROE-mice (see response to comment 5 below). Again, we acknowledge that these findings do not constitute causal proof. Establishing a clear causal relationship would require restoring 4 Hz VTA-PFC synchrony in D2R-OE mice to control levels to see if this rescues the behavioral deficits. However, to the best of our knowledge, such manipulations are currently not technically feasible. This would first require defining the mechanistic underpinnings of dysfunctional synchrony in D2R-OE mice, which we hope our future studies will help to elucidate.

2. Coherence and phase locking both tend to decrease whenever power decreases, simply because it becomes more difficult to estimate the phase of the oscillation. Thus, it is very difficult to interpret the decreases in coherence and phase locking observed here. If the authors could show that correlated changes in the amplitude of the 4 Hz oscillation normally increase from the sample to choice phase, but that this does not occur in D2R-OE mice, this would be better as it should not be subject to the same confound.

Author’s Response: The reviewer raises a very important issue. It is indeed possible that a decrease in 4 Hz power could lead to reduced 4Hz synchrony simply because phase estimates are less accurate when the amplitude of 4 Hz oscillations is smaller. However, it is not clear to what extent decreased power should impair phase estimates in our analyses and we are not aware of any established method for quantifying this. To address this issue we have therefore, as the reviewer suggests, calculated the correlations between 4 Hz power in the VTA and PFC during sample and choice phases in the two genotypes. Consistent with our other measures, we find that 4 Hz VTA-PFC power correlations in control animals are higher in the choice phase, compared to the sample phase. However, this increase is not observed in D2R-OE mice, consistent with coherence and phase-locking measures. These results therefore lend further support to our conclusion that the working memory-dependent recruitment of VTA-PFC coordination is impaired in D2R-OE mice. These results are now described in Supplementary Figure 3 and on p. 6 in the revised manuscript. It is also worth pointing out that the lower 4 Hz power may lead to lower coherence in D2R-OE mice not because phase estimates are less accurate but simply because robust 4 Hz oscillations in the VTA and PFC are necessary for 4

Hz synchrony between the two structures. Thus, VTA-PFC synchrony may be impaired in these animals because they fail to recruit the neural substrate — 4 Hz oscillations — on which this synchrony depends.

3. I'm not sure exactly what to make of the method for estimating number of VTA neurons or VTA neuronal pairs that exhibit significant rhythmicity. Do the authors fit the combined Gaussian / sinusoid to the autocorrelation / cross-correlation, then use the frequency from this fit to construct a sinusoid, and finally test whether that sinusoid is correlated with the auto / cross-correlation? If this is what the authors did, then the p-value should be obtained by bootstrapping, i.e., comparison to appropriately shuffled datasets. Otherwise, I am concerned that the authors may be finding some spurious correlations just because they perform an optimization step (fitting parameters) before testing the correlation. This will automatically induce some degree of correlation between the sinusoid and auto/cross-correlation.

Author's Response: The reviewer's description of our method for estimating rhythmicity is correct. S/he also makes a good point regarding the possibility that it could have led to spurious detection of rhythmic firing, namely that some neurons might have been classified as rhythmic that do not display rhythmic firing. We have therefore modified our method for detecting rhythmicity, along the lines of the bootstrapping method suggested by the reviewer, to take this possibility explicitly into account. In short, the modified method calculates for each correlogram fit the probability that the same or better fit would have been obtained from the correlogram of a non-rhythmic (jittered) spike train. The method first fits the correlograms as before and calculates the goodness-of-fit as the correlation between the sinusoidal component of the fit and the correlogram. The timestamps of the spikes are then jittered to destroy rhythmic firing in the 4 Hz range, and correlograms calculated and fitted in the same way as for the original data. A correlogram is then defined as showing rhythmicity if its goodness-of-fit is greater than the goodness-of-fit of 95% of the correlograms obtained from jittered spike trains, corresponding to a significance threshold of $p < 0.05$. For further details of the method we refer the reviewer to its description in the manuscript on p. 30-31. Using this method, we have re-calculated the % of cells and cell pairs showing rhythmic firing and updated Figure 5 and Supplementary Figure 8 accordingly. Importantly, with this method our findings remain the same as before: The proportion of VTA neurons showing rhythmic 4 Hz firing is not altered in D2R-OE mice whereas the proportion of VTA neuron pairs showing rhythmic co-firing is reduced compared to control mice. We have also confirmed with this method that the number of rhythmically firing neurons is much lower in the PFC than in the VTA. We thank the reviewer for bringing this issue to our attention and hope that her/his concerns regarding this analysis have now been adequately addressed.

4. Overall, it's not clear how the abnormalities in 4 Hz oscillations / synchrony relate to behavioral deficits. In particular, phase-locking of DA neurons to 4 Hz oscillations in either the VTA or PFC is not reduced in D2R-OE mice, it's simply no longer modulated by

task phase (sample vs. choice). Rather than being glossed over, this point should be discussed.

Author's Response: We thank the reviewer for bringing this point to our attention and regret that it was not discussed sufficiently in the manuscript. The reviewer is correct that overall phase-locking of DA neurons to 4 Hz oscillations is not reduced in D2R-OE mice. Specifically, in the original manuscript, we reported that the percentage of VTA DA (as well as GABA and PFC) neurons that were phase-locked to 4 Hz oscillations—when all spikes recorded in a given session were used for analysis—was not lower in D2R-OE mice and that their preferred phase of firing was also similar to control animals (Supplementary Figure 6). The temporal relationship between 4Hz power in the VTA and PFC also showed the same directionality in both genotypes (Supplementary Figure 9). These results suggest that the basic properties of VTA-PFC synchrony may not be fundamentally altered in D2R-OE mice.

Importantly, however, neurons in D2R-OE mice did not display the increase in phase-locking strength from sample to choice that was seen in control animals. (Note that this is not necessarily inconsistent with there not being an overall effect on phase-locking since the spikes recorded in the choice phase during center arm runs are only a small fraction of all spikes). The increase in phase-locking in control animals likely reflects the increased working memory demands of the choice phase (see Fujisawa et al. 2008 for similar results) and the lack of an increase in D2R-OE mice may therefore reflect their failure to recruit the VTA-PFC circuit during working memory performance. This deficit in the task modulation of phase-locking is therefore more selective than a general decrease in phase-locking and, in our opinion, more likely related to the behavioral deficits of D2R-OE mice. Indeed, this phase-locking impairment mirrors the impairment we observed for VTA-PFC 4 Hz coherence (Fig. 3b-d), which correlated with the behavioral deficits of D2R-OE mice (Figure 3e). We therefore conclude that the basic properties of the VTA-PFC circuit are not fundamentally altered in D2R-OE mice but rather its ability to be flexibly modulated and recruited by the WM demands of the task is compromised. We have now discussed this issue on p. 19-20.

5. The authors made a decision to examine the data only after animals had reached criterion for learning. I understand they did this because they want to explore activity in the context of similar behavioral performance. However, this choice makes it hard to determine the relationship of this activity to the behavioral deficits, since the data was not collected in the context of the behavioral deficits. For this reason, it would have been better to analyze data both during and following learning. For example, if the authors found that deficits in rhythmicity / synchrony were not present initially, but developed towards the end of training, then the conclusions / interpretation would change dramatically. Similar, if some of the other properties, e.g., the encoding of L vs. R by PFC and VTA neurons was initially abnormal but then normalized over the course of training, this too would dramatically alter the conclusions.

Author's response: The reviewer raises a very good point. It is correct that we focused our analysis on data obtained after learning to avoid the confounding effects of behavioral differences between the genotypes. Interestingly, we found that although D2R-OE mice performed the task equally well as controls after learning, they nonetheless failed to recruit VTA-PFC synchrony during task performance (i.e. they did not show differences in synchrony between sample and choice). Although these synchrony deficits were seen after the animals had overcome their behavioral deficits (that is, learned the task to the same level as controls), the two deficits could nonetheless be related. Specifically, we hypothesized (p. 7-8) that the failure to recruit the VTA-PFC circuit might contribute to the learning deficits of D2R-OE mice. Supporting this, we found a correlation between the degree of synchrony recruitment during the task and learning impairments in D2R-OE mice. We assumed that a similar synchrony deficit would also be present during the learning stage but other scenarios are also possible including — as suggested by the reviewer — that synchrony is initially normal in D2R-OE mice and that the deficits emerge later. More generally, we agree with the reviewer that examining VTA-PFC synchrony during learning is key for understanding the behavioral deficits of D2R-OE mice.

To address this we have therefore analyzed neural activity recorded during the training phase (first 4 days, corresponding to 40 trials). These results are now presented in Figure 3f and g and on p. 8. We have limited our analysis to LFP coherence because during learning we did not record a sufficient number of neurons and also did not perform our pharmacological test for classifying VTA neurons, which was critical for revealing cell-type specific deficits in the post-learning phase. Interestingly, in control animals we find that the increase in VTA-PFC 4Hz coherence from sample to choice is not present during the initial stage of training (first 10 trials) but emerges later, suggesting a learning effect. (Note that there is a trend for an increase during the first 10 trials; however, this is likely because animals are already performing the task at above-chance levels). In contrast, D2R-OE mice did not display a sample-choice increase in coherence during any of the training days, including the day when they reached criterion performance (Figure 3g). These results therefore argue against the interesting alternative possibility suggested by the reviewer, that synchrony is normal at the beginning of training but later becomes disrupted. Rather, these results suggest that D2R-OE mice fail to recruit the VTA-PFC network already during learning of the task which could account for their learning impairments. A further interesting possibility is that D2R-OE mice may eventually learn and perform the task by recruiting a different brain circuit than controls (see p. 19-20 for discussion). In summary, we believe that these new results from the learning data shed further light on the relationship between behavioral and synchrony deficits in D2R-OE mice and thank the reviewer for his constructive comments.

Reviewer #2 (Remarks to the Author):

Duvarci and colleagues reported rhythmic synchronous activity in PFC and midbrain VTA during a working memory task in wildtype and disease-relevant transgenic (D2R-OE) mice. They found increase of powers and coherences of 4-Hz oscillation in the PFC and

VTA during working memory in wildtype mice but not in D2R-OE mice. The spiking activities of neurons in PFC and VTA were phase-locked to 4-Hz oscillation during the task. Importantly, task-phase-relevant neurons were more strongly modulated by 4-Hz oscillation in wildtype mice, but this was not observed in D2R-OE mice. These results are interesting and timely, and provides new insight into the functional relevance of 4-Hz oscillation in the PFC and VTA network. This study will make a significant contribution to the field, though several issues need to be addressed before publication.

1) One concern is about the relationship between 4-Hz oscillation and behavioral performance. The authors found that the D2R-OE mice had impairment on learning rate (Fig 1b, c) but they performed the task well after they learned it (Fig 1d). This result indicate that 4-Hz oscillation would reflect learning performance. However, recruitment and synchrony of 4-Hz oscillation in the PFC and VTA were only assessed after the mice learned the task (Fig 2 & 3). Since they did recording also in the learning stages (P. 24, Methods), additional analysis of LFP in those periods would more strongly support the link between 4-Hz and behavior.

Author's Response: we are grateful for the reviewer's careful and detailed reading of our study. Although we examined only neural data after learning to ensure that behavioral differences between the D2R-OE mice and controls did not influence the results, we agree with the reviewer that examining the neural data during learning is also important since this is when the behavioral deficits in D2R-OE mice emerge. We have therefore examined VTA-PFC coherence in the sample and choice phases during the initial days of training and now describe these results in Figures 3f and g and on p. 8. We find that in control animals VTA-PFC 4Hz coherence does not differ between sample and choice phases in the early stages of training, but that coherence in the choice phase becomes larger in later stages of training. This suggests that the recruitment of VTA-PFC synchrony in the choice phase develops as animals become better at the task. In contrast, D2R-OE mice fail to show the sample-choice increase in VTA-PFC 4 Hz coherence during training, suggesting that this might underlie their impaired learning of the task. Because reviewer 1 raised the same point we refer reviewer 2 to our response to reviewer 1 (point #5, above) for a more detailed answer.

2) About the result of the correlations between 4-Hz coherence and learning rate in Figure 3e. Here, the authors only showed the result of D2R-OE mice. Is it possible to include the data of control mice?

Author's Response: we did not include the data from the control mice in the original submission because of the low variability in learning rates in this group (all except two animals learned the task in 3-4 days). However, this data is nonetheless informative and we have now included it as Figure 3e (bottom). We have also included data from control animals in supplementary Figure 4. Importantly, we find that control animals display the same relationship between behavior and synchrony as seen in D2R-OE mice: animals that take the longest to learn the task also show the least amount of increase in VTA-PFC coherence from sample to choice ($r=-0.63$, $p<0.01$).

These results thus complement the data from the D2R-OE mice and reinforce the relationship between VTA-PFC synchrony and behavior.

3) To rule out the possibility that the differences of neuronal activities between control and D2R-OE mice were due to motor impairments, quantification of running speeds of mice (or any motor related parameters) would be needed.

Author's Response: We agree with the reviewer that motor impairments are an important potential confound. Although D2R-OE mice have not been found to display gross motor impairments, subtle differences in motor behavior might nonetheless exist that could influence their task performance. To examine this in our dataset we have now compared the running speed of the two genotypes during sessions and task epochs that were included in the analyses, specifically center arm runs during sample and choice phases. These results are now described in Supplementary Figure 2 of the revised manuscript and on p. 7 of the results section. We find that although D2R-OE mice show the same pattern of acceleration and deceleration as they run through the maze, their running speed is on average slightly lower than their control littermates. In order to examine whether these differences in running speed could account for the differences in neural activity that we observed we examined the relationship between running speed and 4 Hz power and coherence in the choice phase on an animal-by-animal basis. To statistically test for the significance of these relationships, we performed an analysis of covariance with genotype as a main factor and running speed as a covariate. This is essential to estimate whether there is a relationship between running speed and neural variables independently of differences between the two genotypes. Importantly, we did not find a significant relationship between running speed and any of these variables. That is, animals with a lower running speed did not display lower 4 Hz power or coherence or smaller changes in these variables between sample and choice. This was also true when we examined correlations separately for each genotype. Running speed was also unrelated to animals' learning rate. The scatter plots examining these relationships and their statistics are all included in Supplementary Figure 2. We conclude from these analyses that although there are differences in running speed between D2R-OE mice and control animals, this cannot account for the differences in neural activity that we observed between the two genotypes.

4) Related above, in the same animals, were the running speeds same between sample and test phases?

Author's Response: This is also a very good point. Supplementary Figure 2 shows the running speed of the two genotypes separately for the sample and choice phases. Running speed did not differ between sample and choice phases for either D2R-OE mice (sample: 32.35 ± 1.16 cm/s; choice: 32.69 ± 1.21 cm/s, $n=17$, $p=0.31$) or control animals (sample: 39.07 ± 1.16 cm/s; choice: 38.51 ± 1.22 cm/s, $n=19$, $p=0.26$). These statistics have now been included in the results section (p. 5) as well as in the figure legend. We therefore conclude that the increases in 4 Hz synchrony that occur in the choice phase are not caused by differences in running speeds in the two task phases.

5) Please provide the explanations of the dotted lines in Fig 6c, d and Figure 7c, f.

Author's Response: the dotted line in Figure 6d represents the chance level for the decoding of task phase. This explanation has been added to the figure legend. Since the decoder attempts to classify task phase into one of two categories ('sample' or 'choice'), chance level is at 50%. For the other figures, the dotted line was meant to represent the false discovery rate for our classification of neurons as phase-selective, turn-selective or reward-responsive. Because this method involved a rank-sum test with a significance level of $p < .05$, we assumed that the false discovery rate is 5%, which is where the dotted line was in these graphs. However, we have now realized that our assumptions about the false discovery rate may not be correct (Colquhoun 2014) and have therefore removed the dotted lines in these figures in the revised manuscript.

Colquhoun, David. 2014. "An Investigation of the False Discovery Rate and the Misinterpretation of P-Values." *Royal Society Open Science* 1 (3):140216.

REVIEWERS' COMMENTS:

Reviewer #1 (Remarks to the Author):

The authors have addressed all of my major concerns. The new data and analyses help to strengthen the manuscript which will be a nice contribution to the field and a new mechanistic lead for future investigation.

Reviewer #2 (Remarks to the Author):

The issues which I raised were fully addressed in the revised manuscript. I do not have further comments.

REVIEWERS' COMMENTS:

Reviewer #1 (Remarks to the Author):

The authors have addressed all of my major concerns. The new data and analyses help to strengthen the manuscript which will be a nice contribution to the field and a new mechanistic lead for future investigation.

Reviewer #2 (Remarks to the Author):

The issues which I raised were fully addressed in the revised manuscript. I do not have further comments.

Author's response: We wish to thank the reviewers once more for their insightful and constructive comments on the manuscript.